# Multilayer relaxor ferroelectric polymer stacks as data transmitter for real-time and programmable infrared information encryption

Yingke Zhu [1,2], Jianghan Wu[1,2], Yang Luo[1], Kede Liu [1], Hyeonji Hong [1], Yuxuan Guo [1], Yuan Meng[1], Meng Gao [1], Hanxiang Wu [1], Jiacheng Fan[1], Yingjie Du[1], Ping He[1] & Qibing Pei [1] ✉

Infrared information encryption is an emerging technique that leverages infrared (IR) radiation for secure data transmission. However, current IR encryption strategies fail to achieve real-time and precise data transmission, increasing the risk of information leakage. Here, utilizing a bistable adhesion polymer as a transfer medium, a multilayer relaxor ferroelectric polymer (RFP) stack is fabricated without additional thermal loads. An 8-layer RFP stack generates a rectangular temperature wave upon the application of an electric field. Its temperature rapidly increases from 22.1 °C to 26.3 °C and remains above 26 °C for over 8 s under an applied electric field of 80 MV/m at 0.01 Hz. Due to the instantaneous electrocaloric effects (temperature change rate up to $10^{-8}$ s/K) upon voltage application, the stack enables a real-time and programmable IR information encryption and decryption strategy. This approach highlights the potential of IR-based communication for secure data transmission, with applications in confidential messaging and optical encryption systems.

Information security is a critical concern across various industries, including communication, military, healthcare, and consumer[1–3]. Infrared (IR) information encryption is an emerging technique that leverages infrared light waves for secure data transfer, providing a promising alternative to traditional encryption methods used in radio frequency (RF) and optical communications[4–6]. According to Stefan–Boltzmann law, the infrared radiation can be regulated by tuning the emissivity or temperature of subject[7]. Researchers have utilized phase-change materials[8,9], thermochromic materials[10–12], electrochromic materials[5,13], and metamaterials[3,14] to manipulate the emissivity. However, emissivity is susceptible to variations due to surrounding environmental conditions and surface contamination[15]. On the other hand, the temperature can be manipulated by

thermoelectric[15,16] or Joule heating[17,18]. However, the heating process typically takes 1 s or more to reach equilibrium and may experience overshoot issues. In addition, Joule heating has the drawback of a low cooling rate due to the slow heat convection to the surrounding environment. Therefore, a precise and real-time infrared manipulation strategy is essential.

The electrocaloric effect (ECE) is a thermodynamic phenomenon where a material undergoes an instant entropy change in response to applied electric field, leading to an adiabatic temperature change ($\Delta T$)[19–23]. The temperature change rate can theoretically reach $10^{-8}$ s/K[24], which makes it a promising candidate for real-time infrared manipulation. However, the temperature change diminishes quickly because of heat dissipation to the environment. Increasing thickness is

[1]Department of Materia Science and Engineering, University of California, Los Angeles, CA, USA. [2]These authors contributed equally: Yingke Zhu, Jianghan Wu. ✉e-mail: qpei@seas.ucla.edu

effective to slow down the temperature decay and boost heat generation[20]. Nevertheless, a thicker film leads to inferior breakdown strength and higher driving voltage[25,26]. Multilayer structure, which combines the thin film's high dielectric strength and the thick film's large thermal mass, seems to be a solution to solve the above concerns. Multilayer ceramic capacitors (MLCC) are widely used in electronic circuits, but only one polymer-based multilayer EC modules were reported[27], where epoxy glue was applied between each layer, which not only increased the passive thermal load but also reduced the overall temperature change ($\Delta T$). Fabricating multilayer ECE polymer films without introducing inactive thermal loads has remained a challenge in the field.

Herein, we report the fabrication of multilayer relaxor ferroelectric polymer (RFP) stacks comprising up to eight layers and free of inactive thermal load by using a bistable adhesion polymer (BAP) as a transfer medium. The adhesive strength of the BAP layer is thermally controlled, enabling it to adhere to and be peeled off from the RFP film to form multilayer stacks. Carbon Nanotube was used as electrodes for its mechanical compliance, good thermal stability, and ease of stacking as it is partially embedded into the surface of the RFP. As the number of

layers increases, the heat generated by the electrocaloric effect (ECE) is amplified, while the heat dissipation rate after the adiabatic temperature change decreases, giving rise to a rectangular-wave temperature profile when a pulsed voltage is applied for a designated duration. For example, the temperature of an 8-layer RFP stack increased from 22.1 °C to 26.3 °C when applying 80 MV/m and remained above 26 °C for up to 8 s. Additionally, it exhibits a rapid response rate of 0.012 s/K. Infrared information encryption and decryption are demonstrated using the 8-layer RFP stack and Morse code. These findings highlight the potential of the multilayer RFP for applications in real-time and accurate infrared information encryption.

## Results

### Multilayer RFP stack fabrication and ECE characterization

Figure 1A shows the schematic fabrication process for the multilayer stack. The stamp or transfer media BAP layer was a thermally activated shape memory polymer with significant modulus change spanning 40–43 °C (Supplementary Fig. 1). The BAP layer was first attached to the RFP film (poly(vinylidene fluoride-trifluoroethylene-chlorofluoroethylene) terpolymer, or P(VDF-TrFE-CFE)) on glass substrate

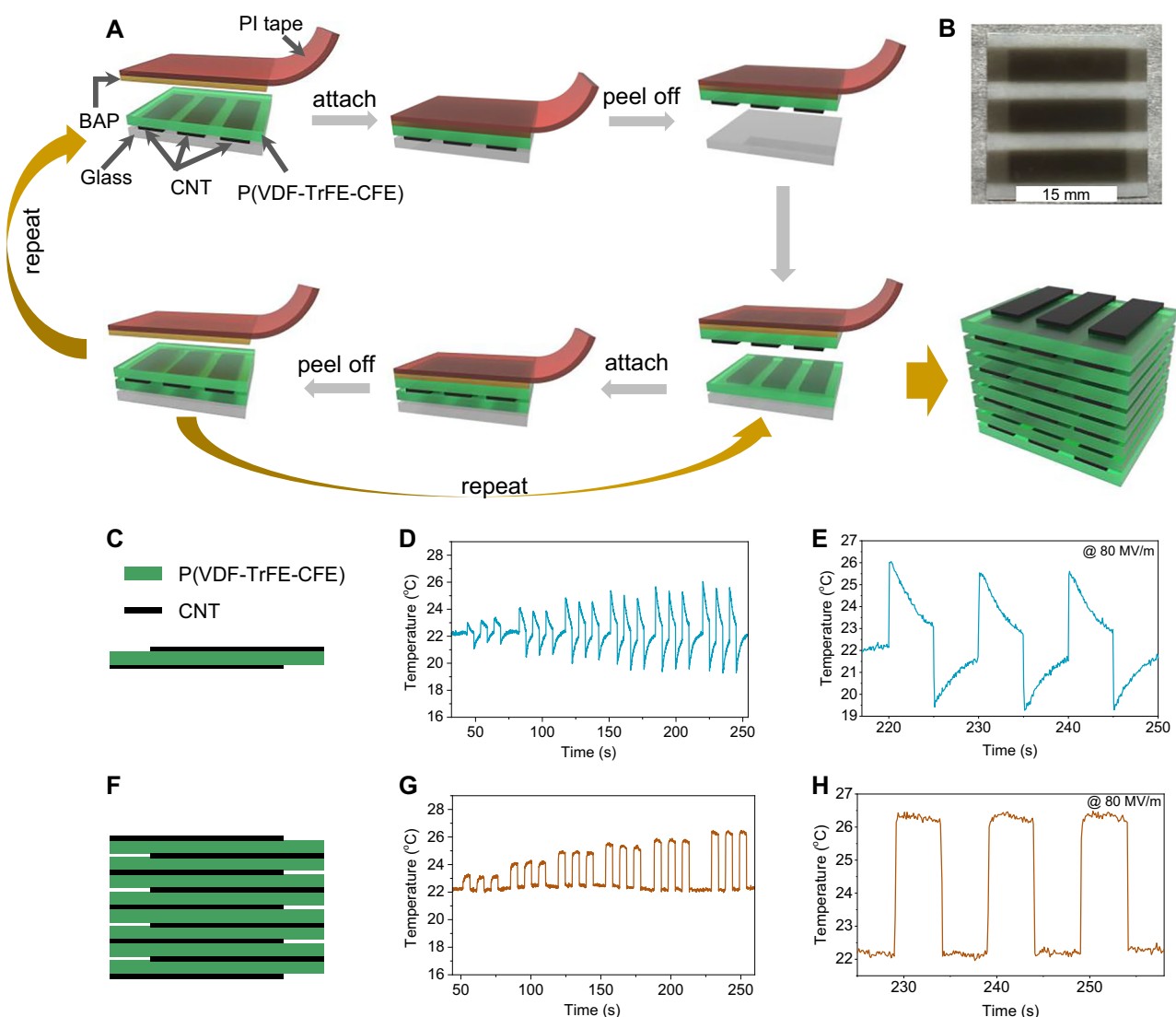

**Fig. 1 | Fabrication diagram of multilayer stack and electrocaloric effect.**
**A** Schematic illustration of the fabrication procedure of a multilayer P(VDF-TrFE-CFE) stack. **B** Optical image of an 8-layer stack. Schematic illustration of **(C)** 1 layer and **(F)** 8-layer stack. Electrocaloric effect of **(D)** 1-layer terpolymer film and **(G)** 8-layer stack from 30 MV/m to 80 MV/m (three tests per electric field). Electrocaloric effect of **(E)** 1-layer film and **(H)** 8-layer stack at 0.1 Hz and 80 MV/m.

(Supplementary Fig. 2a) and then heated up to become sticky. The BAP adhesive layer provided a shape-locking effect, maintaining the terpolymer film's flatness as it was removed. Subsequently, the terpolymer film was transferred onto a terpolymer/glass via heat rolling. After cooling, the BAP adhesive layer could be easily peeled off from the terpolymer, leaving a two-layer stack. This transfer process was repeated until an 8-layer stack was achieved. Figure 1B is the top view image of the stack. As shown in Fig. 1B and Supplementary Fig. 2b, three rectangle active areas (5 mm *17 mm) are formed in one stack. Nine interleaved CNT electrodes were incorporated within the 8-layer stack, forming a multilayer capacitor structure (Fig. 1F). The electrocaloric effect of the 1-layer film (Fig. 1D) and 8-layer stack (Fig. 1G) was tested under an electric field ranging from 30 MV/m to 80 MV/m (Fig. 1E, H), with the terpolymer stack suspended in air. The temperature of 1-layer film exhibits rapid decrease after removing electric field. The temperature of 8-layer stack exhibits little change after the initial rapid temperature rise when electric field is applied or reduction when the field is removed, resulting a rectangular temperature wave (Supplementary Movie 1).

The key to a successful transfer process lies in the ability to control the peel strength at the interfaces between the BAP and terpolymer, as well as between terpolymer layers. Therefore, a 90-degree peeling test (Supplementary Fig. 3A) was conducted to measure the peeling strength between BAP/terpolymer and between terpolymer/terpolymer under different temperatures. As shown in Supplementary Fig. 3B, the peeling strength of the BAP/terpolymer interface at 55 °C (cyan line) is significantly higher than at room temperature (gray line), allowing the terpolymer to be effectively peeled off from substrate by the BAP with the help of water penetration. Meanwhile, the substantial difference in peeling strength between the terpolymer/terpolymer interface (orange line) and the BAP/terpolymer interface (gray line) at room temperature ensures the successful detachment of the BAP after stacking the terpolymer. Additionally, the surface morphology of the terpolymer before (Supplementary Fig. 3C) and after (Supplementary Fig. 3D) contact with BAP reveals a reduction in surface roughness following the stacking process, indicating that BAP does not damage (or roughening) the terpolymer's surface. Furthermore, cross-sectional images of the 8-layer stack (Supplementary Fig. 4) show no cracks or holes, further confirming the integrity of the terpolymer surface after being transferred by BAP.

Figure 2A illustrates the temperature change pattern for stacks ranging from a single layer to eight layers at 0.1 Hz and 60 MV/m. As the number of layers increases, the temperature decay rate following the adiabatic temperature change induced by electric field gradually decreases, ultimately leading to the formation of a rectangular temperature wave in the 8-layer stack. According to Fig. 2B, the $\Delta T$ of P(VDF-TrFE-CFE) stacks are comparable regardless of the number of layers, indicating that the multilayering process does not change the ECE of P(VDF-TrFE-CFE). To further evaluate the temperature maintaining capability of the terpolymer stacks, the ECE of 8-layer stack at 0.01 Hz and 80 MV/m is depicted in Fig. 2C. The stack can withstand 80 MV/m for 50 s, indicating that the stack fabricated using the BAP transfer technique retains the dielectric properties of the terpolymer films. Figure 2D presents a zoomed in view of the green rectangular area in Fig. 2C, showing that the temperature of the 8-layer stack rapidly increases from 22.1 °C to 26.3 °C upon applying an 80 MV/m electric field. Due to the limitation of the framing rate of the IR camera (Supplementary Fig. 5), the measured temperature change rate is limited to 0.012 s/K. When the field is removed, the temperature remains above 26 °C for up to 8 s.

The terpolymer stacks are suspended in air during ECE testing using an IR camera (Supplementary Fig. 6). Therefore, thermal convection to the ambient environment is the primary mechanism of heat dissipation. According to Newton's law of cooling, thermal resistance of convection[28] between terpolymer stack and air can be expressed as

follow:

$$R_{convec} = (T_s - T_a)/q = 1/(hA) \tag{1}$$

where $T_s$ and $T_a$ are the temperature of terpolymer stack and air environment, respectively, $q$ is the dissipated heat from the stack surface, $h$ is the convective heat transfer coefficients, and $A$ is the surface area of the stack. As the thickness of each layer is 50 μm, the $A$ of terpolymer stacks can be calculated by:

$$A = 2*5*17 + 2*0.05*n*17 \tag{2}$$

where n is the layer number of terpolymer stack. Accordingly, the $A$ of a single layer, 2 layers, 3 layers, 4 layers, and 8 layers are 171.7 mm², 173.4 mm², 175.1 mm², 176.8 mm², and 183.6 mm², respectively. The surface area of the 8-layer stack is only 6.9% larger than that of a single layer. However, upon applying an electric field, the heat generated by the 8-layer stack is eight times that of a single layer (Supplementary Fig. 7 and Supplementary Table 1), resulting in a prolonged dissipation time to the environment. We used the thermodynamic model to simulate the temperature decay behavior of the terpolymer stacks upon application and removal of electric field (Supplementary Fig. 8 and Note S1). Simulated temperature decay behavior of an 8-layer RFP stack exhibits good agreement with the experimental results. We also conducted cyclic stability measurement for 8-layer stack (Supplementary Fig. 9), the EC effect retention ratio remains 96% of its initial value after 27,000 cycles under 60 MV/m, demonstrating good cyclic stability for potential practical application.

## IR information encryption and decryption

Based on the rectangular temperature wave of 8-layer RFP stack, an IR information encryption procedure is demonstrated and shown in Fig. 3A. The process begins with encrypting plain text information into corresponding voltage signals using Morse code (Supplementary Fig. 10). These encrypted voltage signals are then applied to an 8-layer FRP stack, generating an IR signal wave that can be captured by an IR camera. Finally, the temperature wave can be decrypted back into plain text. For example, "UCLA" can be encrypted into 4 sets of voltage signals, and then these signals are input into the 8-layer RFP stack (Supplementary Movie 2), the output temperature wave replicates the input voltage signals. Since ECE occurs immediately upon voltage application and the stack exhibits rectangular temperature wave, this demonstration enables real-time and accurate encryption and decryption.

In addition, an 8-layer RFP stack with 7 independently addressable areas were fabricated to display IR images as shown in Fig. 3B. Each active area is independently controlled with in-house designed circuits. The 7-segment panel can display the ten numeric numbers. The numbers "1", "4", and "7" are illustrated in Fig. 3C.

To demonstrate the scalability of our RFP stack, we propose the application of Binary code and Baudot code within the FRP stack, as illustrated in Fig. 4A. For instance, if we consider "0" and "1" in Binary code as representations of short and long electric field waves in the RFP stack, text information can be encrypted using voltage signals and subsequently converted into an IR signal for recognition. In addition, a comprehensive comparison with existing information encryption technologies is presented in Fig. 4B. The "on" and "off" times refer to the information response time upon application and removal of stimulation. Notably, the response time of the RFP stack is several orders of magnitude faster than other encryption technologies.

## Discussion

In summary, multilayer relaxor ferroelectric polymer stacks have been fabricated by using a bistable adhesive polymer as a transfer media. This technique avoids the use of passive thermal loads, the

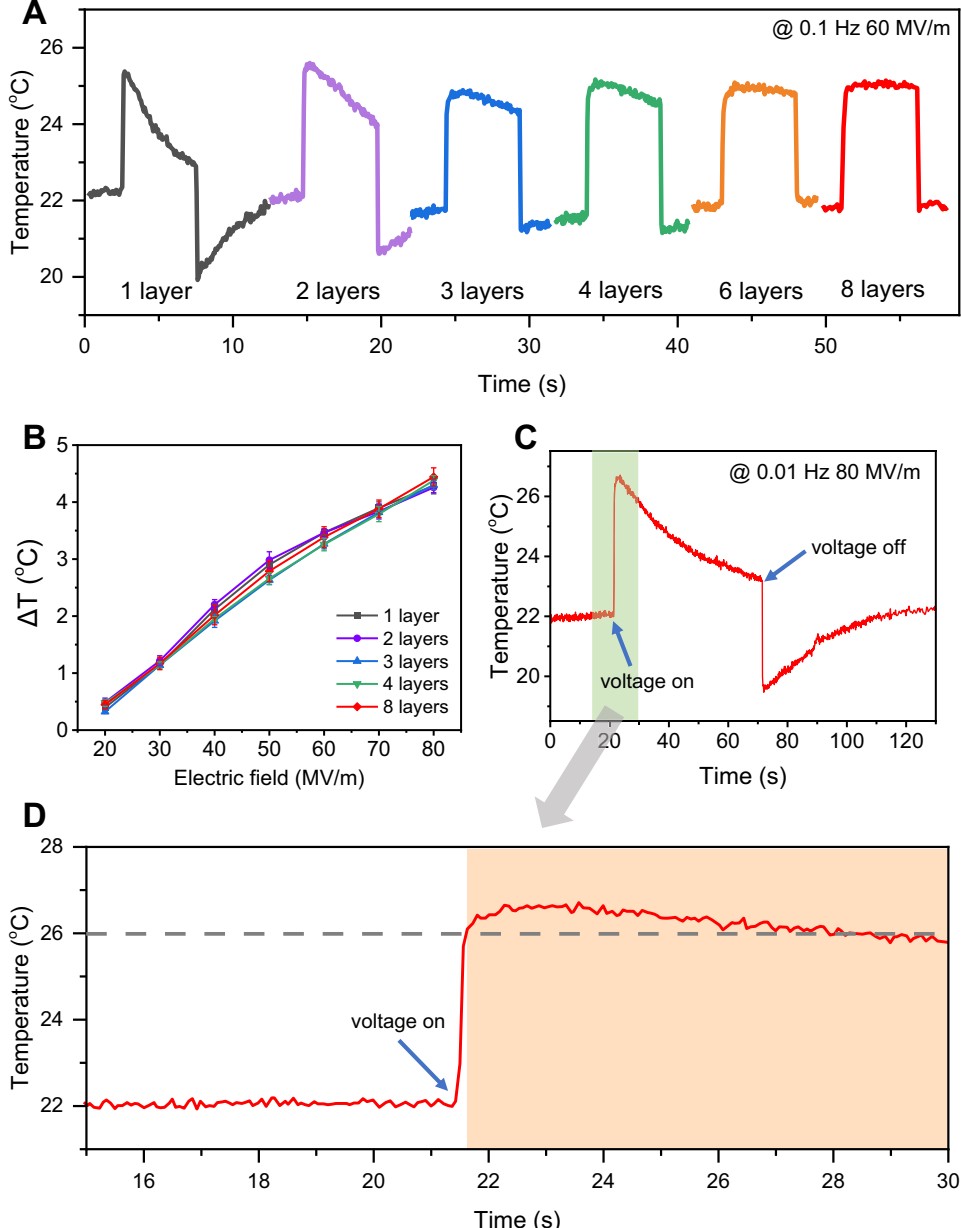

**Fig. 2 | Electrocaloric effects of multilayer stacks. A** Electrocaloric effect of P(VDF-TrFE-CFE) stacks at 0.1 Hz and 60 MV/m. **B** Adiabatic temperature change of the stacks as a function of applied electric field. Data are presented as mean values ± s.d. **C** Electrocaloric effect of the 8-layer stack when applying 80 MV/m at 0.01 Hz and (**D**) enlarged view (insets are infrared images captured before (left) and after (right) applying electric field).

temperature change and heat capacity generated in the stacks due to electrocaloric effect is significantly enhanced compared to a single-layer system. An 8-layer stack exhibits a rectangular temperature wave between 22.1 °C and 26.3 °C under an applied electric field of 80 MV/m at 0.1 Hz, with a rapid temperature response rate of 0.012 s/K. Furthermore, as the multilayer stack is suspended in air, it maintains stable high and low temperatures (<1 °C variation) for approximately 8 s, regardless of the presence of a constant electric field. This precise temperature regulation enables real-time and accurate infrared information encryption and IR display. The response time of the RFP stack is several orders of magnitude faster than existing encryption technologies. In conclusion, this work introduces a promising approach for designing and fabricating multilayer RFP stacks and demonstrating the potential of the stacks for high-speed and programmable information decryption applications.

## Methods

### Materials

P(VDF-TrFE-CFE) (Lot number: 64-020) was purchased from Piezotech Arkema and used as received. N, N-Dimethylformamide (ACS, 99.8%) was purchased from Beantown Chemical. Single-walled carbon nanotubes (SWCNTs, catalog name: P3-SWNT) were purchased from Carbon Solutions, Inc. Urethane diacrylate (UDA, catalog name: CN9021) was obtained from Sartomer and used as received. Stearyl acrylate (SA), trimethylolpropane triacrylate (TMPTA), acrylic acid (AA), 2,2-dimethoxy-2-phenylacetophenone (DMPA), benzophenone (BP), and isopropyl alcohol (IPA) were purchased from Sigma-Aldrich and used as received.

### Bistable adhesive polymer fabrication

The bistable adhesive polymer film was fabricated according to our previously reported procedure[29]. The prepolymer solution

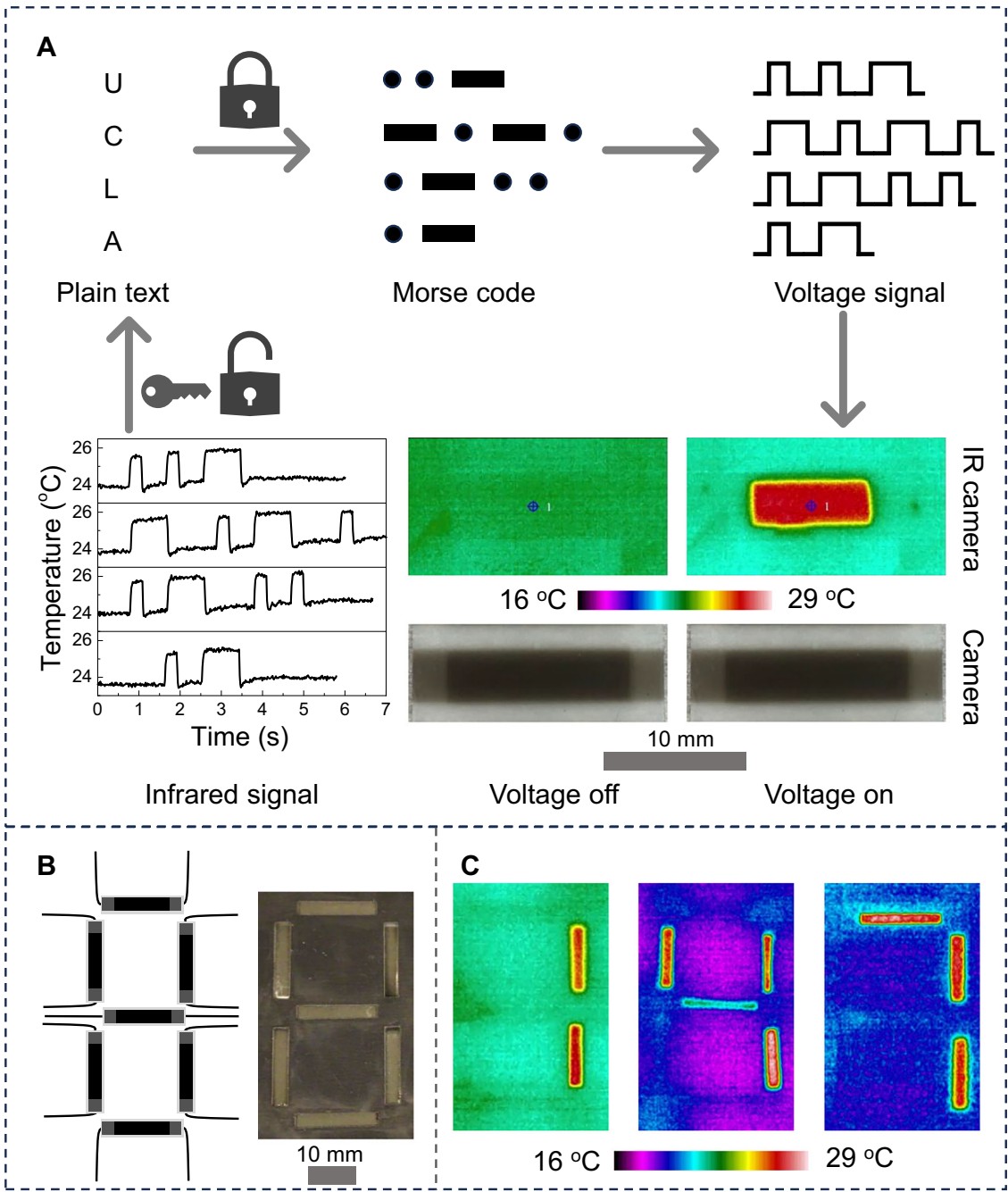

**Fig. 3 | Demonstration of infrared information encryption. A** Schematic of the infrared information encryption procedure. Morse code is used in the algorithm system. Encrypted thermal information "U," C," "L," and "A" is generated by an 8-layer stack. **B** Schematic (left) and optical image (right) of an IR display consisting of 7 pixels formed in an 8-layer stack. **C** Demonstration of number 1, 4, and 7 using IR display.

composition was adjusted to 80 parts (by weight) of SA, 20 parts of UDA, 10 parts of AA, 1.5 parts of TMPTA, 0.5 part of DMPA, and 0.125 part of BP. These components were stirred at 95 °C until no sediment observed and injected between a pair of glass slides on a hot plate with two strips of 50 μm tapes as spacers. Next, the prepolymer was cured through a UV curing conveyor equipped with a Fusion 300S type "H" UV curing bulb for about 3 min. The cured film can be gently peeled off from the glass slides in water.

**Multilayer stack fabrication**
SWCNTs dispersion was prepared by mixing 10 mg P3-SWNT with 18 mL of isopropanol and 2 mL of water. The dispersion was sonicated until no visible aggregation existed. The supernatant was collected

after centrifuging at 8817 g for 15 min. The glass slide covered with PET/PDMS mask was spray-coated with CNT electrodes. P(VDF-TrFE-CFE) powder was dissolved in N, N-Dimethylformamide under vigorous stirring to form a 0.2 g/ml solution. It was then cast on the glass slide by blade coating and dried overnight at 55 °C. The casted terpolymer film was adhered to the BAP by rolling press at 50 °C. After adding water to the corners of the terpolymer film/glass, the capillary action occurs, and water seeps into the interface between the terpolymer film and the glass to weaken the bonding between them; the terpolymer/tape can be easily peeled off from the glass slide. The terpolymer film was further transferred from the tape to another terpolymer film surface by rolling press at 75 °C and then cooled down to peel off from the tape at room temperature. Up to 8 stacks were

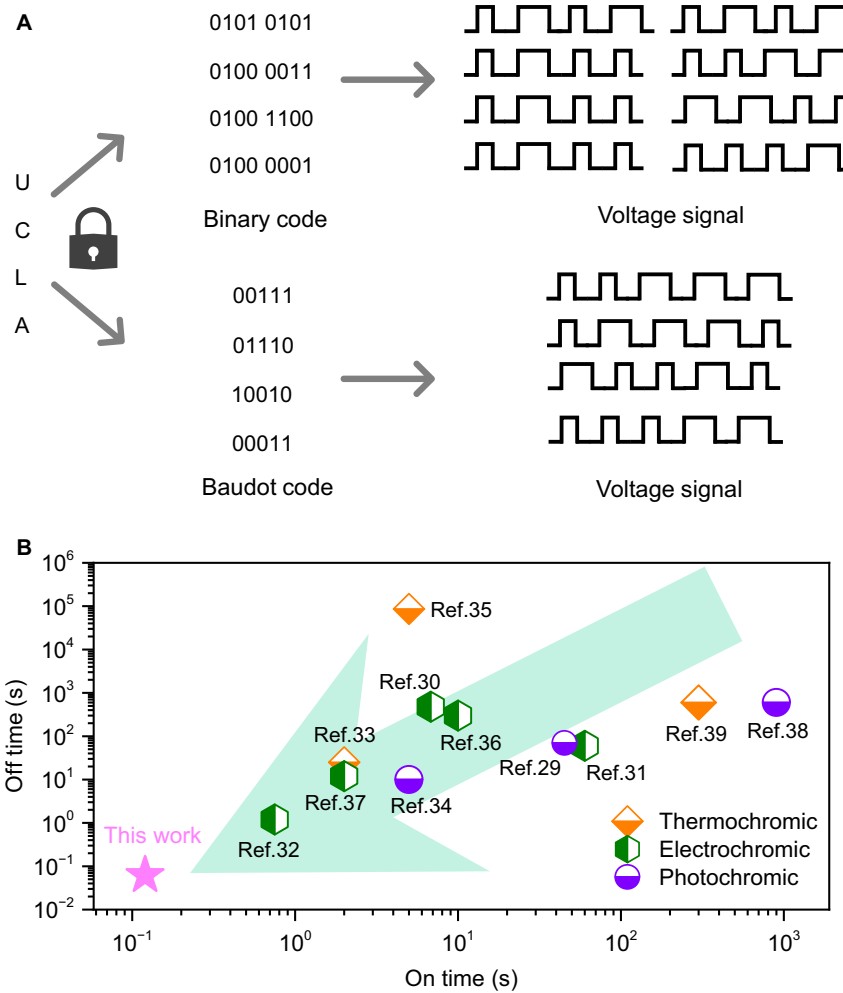

**Fig. 4 | Advantage of RFP stack for information encryption. A** Illustration of codes compatible with the RFP stack. Comparison of encryption speed (**B**) with previous reported information encryption strategies[30–40].

fabricated. The multilayer stack was then annealed in the vacuum oven at 117 °C for around 14 h. The silver paste was applied to the CNT-exposed ends, and the metal wires were attached to the silver paste-covered regions.

### Mechanical characterization

Dynamic temperature sweep tests were conducted with a dynamic mechanical analyzer (TA Instruments RSA) at a temperature ramping rate of 2 °C/min and a frequency of 1 Hz from 25 °C to 55 °C. 90° peeling test was tested using a mechanical testing machine (Univert, CellScale, 50 N load cell, speed: 0.3 mm/s, sample width: 1 inch).

### Electrocaloric characterization

The multilayer stack was connected to a high-voltage power source (10/10B-HS, Trek), and the square wave voltage was applied across the thickness dimension of the stack film. The temperature change due to the electrocaloric effect was recorded using an infrared camera (9320p, ICI Infrared Cameras). A heat flux sensor (HFS-4, OMEGA) was employed and calibrated to measure the heat transferred from the multilayer stack. The multilayer stack was attached to the heat flux sensor with a thin layer of high thermal conductivity paste (OMEGATHERM 201). The thermal flux sensor was placed on an Aluminum heat sink. 8-layer RFP stack was subjected a 60 MV/m electric field to test its cyclic stability.

## Data availability

The data generated in this study are provided in the Supplementary Information/Source Data file, or from the corresponding author upon request. Source data are provided with this paper.

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

## Acknowledgements

The authors acknowledge partial finance support by the Office of Naval Research (award no. N00014-19-1-2212), and the California NanoSystems Institute (CNSI) of the University of California, Los Angeles.

## Author contributions

Y.Z. and Q.P. conceived the idea, and developed the concept. Y.Z., J.W., and Q.P. designed the experiments. Y.Z. and J.W. fabricated RFP stacks and conducted electrocaloric effect measurements. Y.L. helped with the circuit design. K.L. and H.H. helped with the BAP fabrication. Y.G. helped with the electrode connection, Y.D. and P.H. helped with the 90-degree peeling test. Y.M., M.G., W.H., and J.F. offered helpful discussions. Y.Z. and J.W. analyzed the results and prepared the manuscript. Y.Z., J.W., and Q.P. revised and finalized the manuscript with input from all authors.

## Competing interests

The authors declare no competing interests.
