## [Transparent Peer Review file · Nature Communications]

Multilayer Relaxor Ferroelectric Polymer Stacks as data transmitter for Real-time and Programmable Infrared Information Encryption

Corresponding Author: Professor Qibing Pei

Version 0:

Reviewer comments:

Reviewer #1

(Remarks to the Author)

This work presents a multilayer relaxor ferroelectric polymer stack for infrared communication. It exploits the electrocaloric effect to generate rapid temperature change and avoids the fast heat dissipation in monolayer structure with multilayer structure. The results are interesting that more than 4C temperature change is observed, and it can hold over 8 seconds. However, there are serious concerns on this paper.

1.Many of the issues are related with the context of this work.

1a.What are the application scenarios for the developed device? Is it on the personal device, is it on the server, or is it for communication tower? Where this device will be useful? Different application scenarios determine the specs that the device need to meet. For example, the film is 50um in this work, with 80MV/m, it requires 4000V to perform this operation. How feasible is it?

1b.The authors claim that the temperature change method through thermoelectric or joule heating is slow and the electrocaloric is fast. Similarly, what is considered fast or slow. Is 140ms fast enough? It maybe fast compared with the seconds scale approach. But is it fast enough for applications?

1c.It is also unclear how long the temperature need to hold after the field is removed. To the reviewer, holding 8s seems good for temperature, but it also limits the throughput as the next data communication need for the temperature to recover.

1d.How big the temperature change is required? From Fig.3B, this limit will place a lower limit on the electric field to be applied.

Therefore, the authors need to give a context on the application that they are targeting at, which can help the readers appreciate the significance of their results. Also, the authors are suggested to provide benchmarking against the existing solutions and how the proposed solution advances the field.

2.Can the authors also provide arguments on why the P(VDF-TrFE-CFE) film is chosen? There are multiple electrocaloric materials, and then what makes the P(VDF-TrFE-CFE) special?

3.The advantages of IR information encryption are not clear. The analysis shown in Fig.4 is not unique to the IR data transmission. Any communication that requires security will need to have data encrypted before sent out and then decrypted at the receiver side. The device proposed seems like data modulation and transmitter. Is there anything unique related with the device design that can make the encryption better or more secure?

4.How does the electrocaloric effect scale with the film thickness? Can the authors comment on the possible routes on voltage reduction?

5.How is the endurance of the device by repeatedly cycling?

Reviewer #2

(Remarks to the Author)

COMMENTS TO AUTHOR:

Reviewer #1: The manuscript "Multilayer Relaxor Ferroelectric Polymer Stacks for Real-time and Accurate Information Encryption" from Yingke Zhu,* This approach highlights the potential of IR-based communication for secure data transmission. It seems a little interesting. However, the manuscript is mainly focused on the a multilayer P(VDF-TrFE-CFE) stack technology, not on the topic "Infrared Information Encryption". Secondly, the topic of the manuscript is "Information Encryption". In fact, the manuscript is ECE effect of P(VDF-TrFE-CFE) imaged by IR camera. Thirdly, the Morse code is used in 1950-1960s, the type is too old, which can be used in the modern computer system based binary-RAM or magnetic-RAM. Therefore, I can not support to publish the manuscript in high-quality Journal "Nature communication" in the current version. Here are some comments:

1. Why the authors selected 8-layer RFP stack for IR-based communication? How about electrocaloric effect of 5-layer or 6 layer or 7 layer? Please show the optimal layer for these experiments.
2. Figure 2 is about the successful transfer process. I think these parts are suitable to place in the SI, not very important for the main topic of the manuscript.
3. Figure 4A: The infrared information encryption procedure is performed at room temperature. In fact, the procedure is ECE effect of P(VDF-TrFE-CFE). As is well known to all, ECE effect of P(VDF-TrFE-CFE) works well at 30-50 oC. I doubt the working temperature is limited at only 30-40 oC. This devices has obvious shortcomings for data information. Can the varied temperature affect the stability of information signal, for example, at zero temperature, or 50 oC or 100 oC. Can the devices work well?
4. Figure 5C: The numbers "1", "4", and "7" is well shown by IR images. However, the data storage density is too low for information encryption. I think this type of memory is difficult to compatible with modern CMOS platform. One cannot understand how to write/read the memory using electrical circuits fabricated on the basis of modern CMOS platform. Possibly they relied on imaging from IR camera. But these instruments cannot be used for the memory chip.
5. For Information security, the authors do not tell us technical advantages of RFP for Infrared Information Encryption compared with current information storage technology, such as Fe-RAM or magnetic-RAM.

Reviewer #3

(Remarks to the Author)

This paper presents a novel method for infrared information encryption using multilayer relaxor ferroelectric polymer (RFP) stacks. The 8-layer RFP stack generates a rectangular temperature wave when an electric field is applied. The temperature rapidly increases from 22.1°C to 26.3°C and remains above 26°C for over 8 seconds under an electric field of 80 MV/m at 0.01 Hz, enabling real-time and precise infrared information encryption and decryption. However,

1. Were the effects of varying layer counts, material combinations, or other structural parameters on the electrocaloric effect and IR information encryption performance investigated? Could different layer counts or more complex stacking methods beyond the current 8-layer structure further influence performance?
2. The IR information encryption and decryption demo only featured simple text and numbers. It's suggested to conduct experiments on encrypting and decrypting more complex images, videos, or large-scale data. Also, tests on the noise resistance and security of encrypted information, such as accuracy in decryption under noisy or interfering conditions, should be added.
3. It's recommended to carry out long-term stability and durability tests on the multilayer stack. In real - world applications, would environmental factors like temperature fluctuations and humidity changes impact its performance?
4. The paper lacks a comprehensive comparison with existing IR information encryption methods. Besides comparing with single-layer stacks, detailed performance comparisons with other known IR encryption methods (such as those based on thermochromic and electrochromic materials) regarding encryption speed, precision, stability, and energy consumption should be included. This would better define the advantages and limitations of this study and offer readers more complete reference data.
5. For practical use, mass production and long-term stability of multilayer stacks are crucial. The current experiments only showed small-size stack fabrication, without studying the feasibility of mass production. Also, potential performance decline during long-term use wasn't mentioned. It's advised to experiment with larger-size stack fabrication and test its long-term stability and reusability to evaluate the technology's practical feasibility and scalability.

Reviewer #4

(Remarks to the Author)

Version 1:

Reviewer comments:

Reviewer #1

(Remarks to the Author)

The authors have addressed some of the concerns, but still there are many technical challenges of the proposed design.

1. The authors claim that it is feasible to step up the voltage to a few kV from 3.7V supply by citing their previous work. Yes, that can be possible. However, the speed of the voltage conversion limits the speed of the system. There is no discussion on that. Also the authors claim that the theoretical limit for temperature change is 10^{-8} s/K, but the experimentally measured is 0.012s/K. There is a huge gap and there is no discussion on how to approach the theoretical limit. Is it detector limit? If it is, is it even possible to reach that theoretical limit? Overall, there is a lack of insights where this system can be useful for? Is it Kbit/s, Mbit/s, or Gbit/s? Is it suffice for application? There has been efficient hardware for encryption using ASIC at a much faster rate, what new things can this bring?

2. The authors also claim that the stack can return to room temperature instantaneously after removal of the field. This is a very handwaving argument. As this is an important factor impacting the speed of data transmission, the authors need to be quantitative regarding the speed of it.

Reviewer #2

(Remarks to the Author)

The authors have addressed the issue that we concerned. It can be accepted after minor revision. I still suggest the authors provide the endurance of the device by repeatedly cycling.

Reviewer #3

(Remarks to the Author)

Now, the author has basically solved our concerns, and the quality of the manuscript has been greatly improved, meeting the requirements for magazine titles. Therefore, I suggest accepting.

Reviewer #4

(Remarks to the Author)

Version 2:

Reviewer comments:

Reviewer #1

(Remarks to the Author)

[Editor Note: This reviewer provided remarks to the editor and did not list any further technical concerns]

Reviewer #2

(Remarks to the Author)

Accepted as it is

Reviewer #4

(Remarks to the Author)

Response to Reviewers' Comments

In specific response to the points raised by reviewer 1

Comments: This work presents a multilayer relaxor ferroelectric polymer stack for infrared communication. It exploits the electrocaloric effect to generate rapid temperature change and avoids the fast heat dissipation in monolayer structure with multilayer structure. The results are interesting that more than 4°C temperature change is observed, and it can hold over 8 seconds. However, there are serious concerns on this paper.

Remarks 1: What are the application scenarios for the developed device? Is it on the personal device, is it on the server, or is it for communication tower? Where this device will be useful? Different application scenarios determine the specs that the device need to meet. For example, the film is 50µm in this work, with 80MV/m, it requires 4000V to perform this operation. How feasible is it?

Response 1: Thank you very much for raising these good questions. Our relaxor ferroelectric polymer stack can function as a data modulator for real-time and programmable infrared (IR) information encryption. This makes our stack well-suited for applications requiring secure IR communication. For instance, it can be integrated into personal devices for potential real-time encrypted IR transmission. Regarding the driving voltage, our stack demonstrates a 4 °C temperature change under an electric field of 80 MV/m. As shown in Figure 2B of the revised manuscript, a 2 °C temperature change is achieved at 40 MV/m. This indicates that high electric fields are not strictly necessary for practical applications, provided the IR camera can accurately detect the modulation.

Additionally, in our previous work (Science Advances, 10, 43, 2024, DOI: 10.1126/sciadv.adr176), we designed a circuit using a DC/DC converter to step up a 3.7 V DC input to 4 kV. This output was used to drive nine pairs of commercial miniature HV optocouplers for switching. In the current manuscript, we employed the same circuit to achieve number display in Figure 3. With IRB board approval, we conducted human subject test of haptic devices which were applied to the users' palms. Despite the high V, the current is one the order microA, and circuit can be driven with small batteries. Therefore, we believe the high driving voltage is not a major issue in practical applications.

Remarks 2: The authors claim that the temperature change method through thermoelectric or joule heating is slow and the electrocaloric is fast. Similarly, what is considered fast or slow. Is 140ms fast enough? It maybe fast compared with the seconds scale approach. But is it fast enough for applications?

Response 2: Thank you very much for your question. As outlined in the Introduction, the electrocaloric effect (ECE) theoretically allows for a temperature change rate of up to 10^{-8} s/K. However, in our experiments, the observed temperature change rate occurred over approximately 0.012 s/K, which we attribute to the limited frame rate of the IR camera used. Despite this limitation, the results demonstrate that our device is capable of achieving real-time and on-demand IR information encryption.

Remarks 3: It is also unclear how long the temperature need to hold after the field is removed. To the reviewer, holding 8s seems good for temperature, but it also limits the throughput as the next data communication need for the temperature to recover.

Response 3: The key advantage of the multilayer stack is its ability to regulate temperature through frequency and electric field adjustments, maintaining it between room temperature and a specific threshold to meet data transmission requirements. The maximum temperature holding time is 8 seconds, though continuous operation at this duration is unnecessary. Upon removal of the electric field, the stack's temperature instantaneously returns to room temperature.

Remarks 4: How big the temperature change is required? From Fig.3B, this limit will place a lower limit on the electric field to be applied.

Response 4: We think that the ΔT requirement depends on the accuracy of the IR camera or algorithm in certain applications. For example, if the noisy resistance of IR camera is low, it might

need higher temperature change to ensure the accuracy of data transmission.

Remarks 5: Can the authors also provide arguments on why the P(VDF-TrFE-CFE) film is chosen? There are multiple electrocaloric materials, and then what makes the P(VDF-TrFE-CFE) special?

Response 5: Compared to other polymers and inorganic electrocaloric materials, P(VDF-TrFE-CFE) offers advantages including a giant electrocaloric effect near room temperature and inherent flexibility, making it well-suited for wearable and flexible electronic devices.

Remarks 6: The advantages of IR information encryption are not clear. The analysis shown in Fig.4 is not unique to the IR data transmission. Any communication that requires security will need to have data encrypted before sent out and then decrypted at the receiver side. The device proposed seems like data modulation and transmitter. Is there anything unique related with the device design that can make the encryption better or more secure?

Response 6: According to the suggestion offered by the reviewer, we have changed the title of our manuscript to “*Multilayer Relaxor Ferroelectric Polymer Stacks as data transmitter for Real-time and Programmable Infrared Information Encryption*”. The key advantage of our multilayer stack is its ability to regulate temperature through frequency and electric field adjustments, maintaining it between room temperature and a specific threshold to meet data transmission requirements, which has never been reported before. The temperature change rate of electrocaloric effect can be as high as 10^8 K/s, which makes our stack possess unique temperature regulation capability. Therefore, the unique feature of our device is its unprecedented temperature regulation capability, enabling real-time and on-demand IR communication.

Remarks 7: How does the electrocaloric effect scale with the film thickness? Can the authors comment on the possible routes on voltage reduction?

Response 7: The electrocaloric (EC) effect is primarily driven by the entropy change that occurs when an electric field is applied or removed. As such, thinner films would exhibit the EC effect as thicker films. However, to obtain the same heat capacity, one would need to stack more layers of thinner films. In terms of voltage reduction to the tens or a few volts, we previously developed a circuit using a DC/DC converter to transform a 3.7 V DC input into 4 kV (Science Advances, 10, 43, 2024, DOI: 10.1126/sciadv.adr176). This voltage was then directed to nine pairs of commercial miniature HV optocouplers for switching. In this manuscript, we also utilized this circuit for number display in Figure 3, demonstrating its versatility across different applications. Therefore, we can use a converter to solve the high driving voltage issue.

Remarks 8: How is the endurance of the device by repeatedly cycling?

Response 8: Our previous work (Science, 386, 546–551, 2024) showed that the P(VDF-TrFE-CFE) can survive 36000 cycles under 70 MV/m and 1 Hz. Since this work used the same material and coating protocol, we believe that they should share similar endurance.

In specific response to the points raised by reviewer 2

Comments: The manuscript "Multilayer Relaxor Ferroelectric Polymer Stacks for Real-time and Accurate Information Encryption" from Yingke Zhu,* This approach highlights the potential of IR-based communication for secure data transmission. It seems a little interesting. However, the manuscript is mainly focused on the a multilayer P(VDF-TrFE-CFE) stack technology, not on the topic “Infrared Information Encryption”. Secondly, the topic of the manuscript is “Information Encryption”. In fact, the manuscript is ECE effect of P(VDF-TrFE-CFE) imaged by IR camera. Thirdly, the Morse code is used in 1950-1960s, the type is too old, which can be used in the modern computer system based binary-RAM or magnetic-RAM. Therefore, I can not support to publish the manuscript in high-quality Journal “Nature communication” in the current version.

Remarks 1: Why the authors selected 8-layer RFP stack for IR-based communication? How about electrocaloric effect of 5-layer or 6 layer or 7 layer? Please show the optimal layer for these experiments.

Response 1: Thank you very much for these good questions. The EC effect of 6-layer stack under 60 MV/m was updated Figure 2A in the modified manuscript. One can see that there is a temperature drop upon applying electric field. In order to ensure a good accuracy, we choose 8-layer RFP stack because it can hold temperature up to 8s, thus making it a promising candidate for real-time and on-demand IR information encryption. Also note that stacked devices generally use even number of layers such that the two outer electrodes are grounded.

Remarks 2: Figure 2 is about the successful transfer process. I think these parts are suitable to place in the SI, not very important for the main topic of the manuscript.

Response 2: Figure 2 has been moved to the supporting information (Figure S3).

Remarks 3: Figure 4A: The infrared information encryption procedure is performed at room temperature. In fact, the procedure is ECE effect of P(VDF-TrFE-CFE). As is well known to all, ECE effect of P(VDF-TrFE-CFE) works well at 30-50 °C. I doubt the working temperature is limited at only 30-40 °C. This device has obvious shortcomings for data information. Can the varied temperature affect the stability of information signal, for example, at zero temperature, or 50 °C or 100 °C. Can the devices work well?

Response 3: P(VDF-TrFE-CFE) exhibits a giant electrocaloric effect near room temperature, suggesting an optimal operating range between 5° C and 50° C (Appl. Phys. Lett. 99, 052907, 2011). For applications requiring higher temperatures, we believe our strategy can also be applied to fabricate multilayer P(VDF-TrFE) stacks, which exhibit electrocaloric effects between 80° C and 110° C (J. Mater. Chem. C, 2013, 1, 23 – 37).

Remarks 4: Figure 5C: The numbers “1”, “4”, and “7” is well shown by IR images. However, the data storage density is too low for information encryption. I think this type of memory is difficult to compatible with modern CMOS platform. One cannot understand how to write/read the memory using electrical circuits fabricated on the basis of modern CMOS platform. Possibly they relied on imaging from IR camera. But these instruments cannot be used for the memory chip.

Response 4: First, I would like to clarify that our RFP stack is able to work as a data transmitter for real-time and on-demand IR information encryption. Figure 5C is only one of the demonstrations that the RFP stack can do. The temperature of our RFP stack rapidly increases from 22.1° C to 26.3° C and remains above 26° C for over 8 seconds under an applied electric field of 80 MV/m at 0.01 Hz. Such unprecedented temperature manipulation capability is never seen in previous reported literatures, which makes it promising in real-time and precise IR information encryption and decryption strategy.

Remarks 5: For Information security, the authors do not tell us technical advantages of RFP for Infrared Information Encryption compared with current information storage technology, such as Fe-RAM or magnetic-RAM.

Response 5: I would like to clarify that the RFP stack in this manuscript works as a data transmitter for IR information encryption, not as information storage technology. Therefore, it cannot be compared with Fe-RAM or magnetic-RAM.

In specific response to the points raised by reviewer 3

Comments: This paper presents a novel method for infrared information encryption using multilayer relaxor ferroelectric polymer (RFP) stacks. The 8-layer RFP stack generates a rectangular temperature wave when an electric field is applied. The temperature rapidly increases from 22.1°C to 26.3°C and

remains above 26°C for over 8 seconds under an electric field of 80 MV/m at 0.01 Hz, enabling real-time and precise infrared information encryption and decryption.

Remarks 1: Were the effects of varying layer counts, material combinations, or other structural parameters on the electrocaloric effect and IR information encryption performance investigated? Could different layer counts or more complex stacking methods beyond the current 8-layer structure further influence performance?

Response 1: Thank you very much for bringing up these questions. We investigated the number of layers on the electrocaloric effect of RFP stacks, as shown in Figure 2A. We found that with increasing number of layers, the EC effect of RFP stacks gradually changes to a rectangular wave because of the heat dissipation rate become slower. However, these changes will not affect the temperature change rate because the theoretically temperature change rate of electrocaloric effect (ECE) is as high as 10^{-8} s/K. If the layer number is further increased, the maximum temperature holding time upon applying electric field will also increase.

Remarks 2: The IR information encryption and decryption demo only featured simple text and numbers. It's suggested to conduct experiments on encrypting and decrypting more complex images, videos, or large-scale data. Also, tests on the noise resistance and security of encrypted information, such as accuracy in decryption under noisy or interfering conditions, should be added.

Response 2: The unique function of the stack in our work is to serve as a data transmitter that converts electrical signals into thermal signals in real time. Its key advantage over other information encryption strategies lies in leveraging the electrocaloric effect, which enables an instantaneous temperature change (up to 10^{-8} s/K). In our previous study (Science, 386, 546–551, 2024), P(VDF-TrFE-CFE) demonstrated durability over 36,000 cycles under conditions of 70 MV/m and 1 Hz. Since the same material batch is used in this work, we expect comparable endurance. This robustness allows our technology to support large-scale data encryption and decryption. Furthermore, because the electrocaloric effect is governed solely by entropy change (Nature, 600, 7890, 664–669, 2021), the encryption process remains unaffected by external noise, ensuring high accuracy.

Remarks 3: It's recommended to carry out long-term stability and durability tests on the multilayer stack. In real - world applications, would environmental factors like temperature fluctuations and humidity changes impact its performance?

Response 3: In our previous work (Science, 386, 546–551, 2024), the P(VDF-TrFE-CFE) can survive 36000 cycles under 70 MV/m and 1 Hz. Since this work used the same batch, the stack is expected to exhibit similar endurance. In terms of temperature fluctuations, because the Curie transition temperature of P(VDF-TrFE-CFE) is around room temperature, previous result show that it can work well under 5 to 50 °C (Appl. Phys. Lett. 99, 052907, 2011). Also, while environmental temperature change is slow, the EC effect is instantaneous, allowing for accurate data transmission. Regarding humidity impact, EC effect measurements were done in our lab without humidity control. The P(VDF-TrFE-CFE) works well in the environmental humidity range from 10 to 70%.

Remarks 4: The paper lacks a comprehensive comparison with existing IR information encryption methods. Besides comparing with single-layer stacks, detailed performance comparisons with other known IR encryption methods (such as those based on thermochromic and electrochromic materials) regarding encryption speed, precision, stability, and energy consumption should be included. This would better define the advantages and limitations of this study and offer readers more complete reference data.

Response 4: Figure 4B has been added in the revised manuscript for a comprehensive comparison with existing information encryption strategies. Notably, the response time of our RFP stack is several orders of magnitude faster than other encryption technologies, underscoring its superior performance. A shorter response time allows the system to complete data transmission or encryption/decryption

more rapidly, thereby minimizing the time window available for potential interception or signal analysis. This characteristic makes our RFP stack promising for real-time, high-security information encryption applications.

Figure 4B. Comparison of encryption speed (B) with previous reported information encryption strategies.

Remarks 5: For practical use, mass production and long-term stability of multilayer stacks are crucial. The current experiments only showed small-size stack fabrication, without studying the feasibility of mass production. Also, potential performance decline during long-term use wasn't mentioned. It's advised to experiment with larger-size stack fabrication and test its long-term stability and reusability to evaluate the technology's practical feasibility and scalability.

Response 5: As shown in Figure S2, we were able to make three stacks in one batch with each stack size of 5 mm*17 mm, which we believe is sufficient for current applications. Coating larger-area films is not a major issue for soluble polymers. In terms of the long-term stability, our previous work (Science, 386, 546–551, 2024) tested P(VDF-TrFE-CFE) films to 36000 cycles under 70 MV/m and 1 Hz. Since this work used the same batch of material and same coating protocol, the stacked devices are expected to similar cycle lifetime.

In specific response to the points raised by reviewer 1

Comments: The authors have addressed some of the concerns, but still there are many technical challenges of the proposed design.

Remarks 1: The authors claim that it is feasible to step up the voltage to a few kV from 3.7V supply by citing their previous work. Yes, that can be possible. However, the speed of the voltage conversion limits the speed of the system. There is no discussion on that. Also the authors claim that the theoretical limit for temperature change is 10^{-8} s/K, but the experimentally measured is 0.012s/K. There is a huge gap and there is no discussion on how to approach the theoretical limit. Is it detector limit? If it is, is it even possible to reach that theoretical limit? Overall, there is a lack of insights where this system can be useful for? Is it Kbit/s, Mbit/s, or Gbit/s? Is it suffice for application? There has been efficient hardware for encryption using ASIC at a much faster rate, what new things can this bring?

Response 1: Thank you very much for raising these good questions. First, in our previous work (Science Advances, 2024, 10, 43, DOI: 10.1126/sciadv.adr176), HV optocouplers (model OPTO-150, HVM Technology; turn-on/turn-off time = 2 μ s) were used for switching in the high-voltage power supply. The voltage conversion speed is mainly determined by the power capacity of the optocouplers and the capacitance of the stack. This discussion has been added to the Supplementary Information and highlighted in red.

Regarding the temperature rate, the IR camera used in this work operates at a frame rate of 60 Hz (Figure S5), which constrains the experimental results presented in our manuscript. We have discussed this limitation in the manuscript and highlighted it in blue. Achieving the theoretical electrocaloric (EC) temperature rate of 10^{-8} s /K would require an IR camera with a significantly higher frame rate. For example, in 2009 (*Nature*, 458, 1145–1149), Keisuke Goda proposed a Serial Time-Encoded Amplified Microscopy technique that achieved frame rates up to 6.1 MHz. More recently, in 2021 (*Photonics*, 8(2), 34), Amir Matin reported a specialized system capable of reaching 20 GHz using compressive sensing principles. It should be able to transfer data with a speed between Mbit/s and Gbit/s if the frame rate can reach the theoretical limit.

ASICs are efficient hardware solutions for encryption, but they also come with notable disadvantages, such as limited flexibility and a high risk of obsolescence. Designed for a single, fixed function, ASICs cannot be easily reprogrammed for other purposes, making them unsuitable for applications that require frequent algorithm updates or modifications. Consequently, if standards or encryption protocols change, an ASIC may become unusable because it cannot be upgraded. In contrast, our work leverages the electrocaloric effect to enable a stack that functions as a data transmitter capable of real-time, on-demand, and programmable infrared information encryption, offering a level of adaptability and future-proofing that surpasses conventional ASIC-based solutions.

Remarks 2: The authors also claim that the stack can return to room temperature instantaneously after removal of the field. This is a very handwaving argument. As this is an important factor impacting the speed of data transmission, the authors need to be quantitative regarding the speed of it.

Response 2: Thank you very much for your suggestion. Due to the frame rate limitation of our IR camera, the measured temperature change rate is 0.012 s/K. As discussed above, a significantly higher rate could be captured using higher-frame-rate IR cameras. Additionally, we have included in the abstract (highlighted in red) a theoretical temperature change rate of 10^{-8} s/K.

In specific response to the points raised by reviewer 2

Comments: The authors have addressed the issue that we concerned. It can be accepted after minor revision. I still suggest the authors provide the endurance of the device by repeatedly cycling.

Response: Thank you very much for your suggestion. We tested the endurance of 8-layer stack device by repeatedly cycling under 60 MV/m. As shown in Figure S9, the EC effect of 8-layer stack device remains 96% or its initial value after 27000 cycles, exhibiting great cycling stability.

Figure S9. Cyclic performance of 8-layer stack under 60 MV/m.